# Vaccine Coverage against Influenza and SARS-CoV-2 in Health Sciences Students during COVID-19 Pandemic in Spain

**DOI:** 10.3390/vaccines10020159

**Published:** 2022-01-21

**Authors:** María Julia Ajejas Bazán, Ramón Del Gallego-Lastra, Cristina Maria Alves Marques-Vieira, Candelas López-López, Silvia Domínguez-Fernández, Milagros Rico-Blázquez, Francisco Javier Pérez-Rivas

**Affiliations:** 1Departamento de Enfermería, Facultad de Enfermería, Fisioterapia y Podología, Universidad Complutense de Madrid, 28040 Madrid, Spain; rgallego@ucm.es (R.D.G.-L.); canlopez@ucm.es (C.L.-L.); milarico@ucm.es (M.R.-B.); frjperez@ucm.es (F.J.P.-R.); 2Academia Central de la Defensa, Escuela Militar de Sanidad, Ministerio de Defensa, 28040 Madrid, Spain; 3Grupo de Investigación UCM “Salud Pública-Estilos de Vida, Metodología Enfermera y Cuidados en el Entorno Comunitario”, Departamento de Enfermería, Facultad de Enfermería, Fisioterapia y Podología, Universidad Complutense de Madrid, 28040 Madrid, Spain; sildom01@ucm.es; 4Grupo de Investigación UCM “Humanidades, Ciencia y Salud”, Departamento de Enfermería, Facultad de Enfermería, Fisioterapia y Podología, Universidad Complutense de Madrid, 28035 Madrid, Spain; 5Center Interdisciplinary Research in Health (CIIS), Nursing School (Lisbon), Institute of Health Sciences, Universidade Católica Portuguesa, 1649-023 Lisbon, Portugal; cristina_marques@ucp.pt; 6Department of Intensive Care, Hospital Universitario 12 de Octubre, 28041 Madrid, Spain; 7Grupo de Investigación en Cuidados (InveCuid), Instituto de Investigación Sanitaria Hospital 12 de Octubre (imas12), 28041 Madrid, Spain; 8Centro Municipal de Salud Comunitaria Centro, Madrid Salud, Ayuntamiento de Madrid, 28013 Madrid, Spain; 9Unidad de Investigación de la Gerencia Asistencial de Atención Primaria, Servicio Madrileño de la Salud, 28035 Madrid, Spain

**Keywords:** COVID-19 vaccines, vaccination coverage, influenza vaccines, students, health occupation

## Abstract

Vaccination against influenza and SARS-CoV-2 is recommended in health sciences students to reduce the risk of acquiring these diseases and transmitting them to patients. The aim of the study was to evaluate how the pandemic influenced the modification of influenza vaccination coverage during the 2019/2020 and 2021/2022 campaigns and to analyze the vaccination coverage against SARS-CoV-2 in health sciences students. A cross-sectional study was conducted among students of the Faculty of Nursing, Physiotherapy and Podiatry of the Complutense University of Madrid. A questionnaire was administered in two stages, the first, Q1, before the start of the pandemic, where we analyzed influenza coverage during the 2019/2020 campaign and a second, Q2, 18 months after the start of the pandemic where we analyzed influenza coverage during the 2021/2022 campaign and coverage against SARS-CoV-2. A total of 1894 students (58.78% of the total of those enrolled) participated. Flu vaccination coverage increased from 26.7% in Q1 to 35.0% in Q2 (*p* < 0.05), being higher in the age group older than 21 years, who studied nursing, were in their fourth year and lived with people at risk. Vaccination coverage against SARS-CoV-2 was very high (97.8%), especially in students vaccinated against influenza. Coverage of the influenza vaccine in health sciences students increased from 2019–2020 to 2021–2022, being higher in the age group older than 21 years, who studied nursing, were in their first and fourth year and lived with people at risk. Coverage of the SARS-CoV-2 vaccine in health sciences students was very high, especially in those vaccinated against influenza.

## 1. Introduction

Vaccination has sufficient evidence to be considered one of the most effective public health interventions to control communicable diseases. Despite this, there are anti-vaccine movements that, in recent years and during the COVID-19 pandemic, have led to a growing rejection of vaccination [1,2].

The role played by health professionals with regard to doubts about vaccines is decisive [3,4]. They are the main source of information for patients on immunization and are often trusted by them [5,6]. Likewise, at the end of their studies, health sciences students will become health professionals who, in order to carry out their professional activity in a responsible manner, must be aware of the importance of being vaccinated against influenza and SARS-CoV-2 to protect patients and people close to them. The pandemic caused by COVID-19 has left a high percentage of morbidity and mortality as well as economic costs worldwide and in Spain [7]. Currently, there is still no clear treatment of the disease, and the mass vaccination campaigns initiated at the end of 2020 are one of the most effective measures to reduce the health and socioeconomic impacts of the pandemic. As a consequence of this massive campaign launched worldwide, there has been a change in the general confidence in vaccines and the willingness to receive the next dose of influenza vaccine, increasing from 53% to 60% [8]. Both the perception of risk associated with influenza and preventive behaviors may change, causing a relative increase in people’s intention to be vaccinated of about 10% (from 44.1% in 2020 to 48.6% in 2021 surveys). However, it is unclear whether this increased willingness will translate into higher coverage [9]. It is also important to highlight the increase in influenza vaccination coverage in the 2020/2021 season in different population groups in many European Countries [10] including Spain, despite the low influenza activity or the existence of a problem of under-diagnosis, when thinking that COVID 19 cases could not suffer simultaneously from both pathologies. Currently, very few studies are available on changes in influenza coverage. In Spain, students of health science are recommended, by the University, to be vaccinated against influenza and, currently, against SARS-CoV-2 before starting their hospital internships. They are vaccinated at their health centers by appointment requested by them. The administration of these two vaccinations is free of charge for the students.

The recommendations for influenza vaccination for the 2021–2022 campaign, due to the synergistic effect observed in the case of co-infection with SARSCoV-2, which doubles the risk of death [11,12,13], place special emphasis on increasing influenza vaccination coverage, especially in at risk groups, including trainees in healthcare and social-healthcare settings. The aim is to protect those most vulnerable to complications and those who can transmit the disease to others. 

Prior to this study, another investigation was carried out to examine the general attitude towards vaccination in students of the Faculty of Nursing, Physiotherapy and Podiatry of a university in Madrid, Spain, using the “Questionnaire of Attitudes and Behavior towards Vaccination of Health sciences Students of Health sciences,” and the rate of influenza vaccination coverage was calculated before the onset of the pandemic. The objectives of this study were to evaluate the change in influenza vaccination coverage during the 2019/2020 and 2021/2022 campaigns, to analyze the current vaccination coverage against SARS-CoV-2, to identify factors associated with vaccination, in both cases, in health sciences students and to describe the severity of the signs and symptoms presented by the students when they were diagnosed with a COVID 19 case. 

## 2. Materials and Methods

A cross-sectional study was conducted among students of the Faculty of Nursing, Physiotherapy and Podiatry, Complutense University of Madrid, who agreed to participate and gave their informed consent. This faculty was the only one chosen because of the accessibility of the researchers and the management of resources, pending the results obtained. As an initial hypothesis, it was assumed that vaccination coverage would increase considerably as a consequence of the pandemic. The study subjects were students enrolled for the year 2019/2020, i.e., before the start of the pandemic, and those enrolled for the year 2021/22, i.e., 18 months after the pandemic was declared. All students of the faculty were included, without any type of sampling. Data collection took place between 15–30 January 2020 (Q1), the 2019/2020 campaign and during October to November, 2021 (Q2), where data were collected for the 2021/2022 campaign that was underway. Most of the students had already been vaccinated and those missing had an appointment for vaccination, the latter being counted as vaccinated. In no case was reference made to the intention to vaccinate.

A total of 1582 students were included in Q1 from the nursing, 1039 (65.6%), from the physiotherapy, 262 (16.6%), and from the podiatry degrees, 281 (17.8%). In Q2, 1640 students were included, from nursing 1024 (62.4%), physiotherapy 300 (18.3%) and podiatry 316 (19.3%). The students who participated were not the same. Those from two nursing, physiotherapy and podiatry courses coincided, the rest were different.

All enrolled students were asked to complete a questionnaire on vaccination online which included the following variables: sex, age (<21 years old, ≥21 years; 21 years of age was chosen as the limit in the categorization of the variable age, as it was the value at which the number of students of this age and above decreased), degree being studied at the time (Nursing, Physiotherapy or Podiatry), year (first, second, third or fourth) and whether they had received the flu vaccine in the last year (yes, no). The vaccination questionnaire given to the students who participated in Q2 also included variables related to COVID-19: having received the SARS-CoV-2 vaccine (yes, no), having been a confirmed case of COVID-19 (yes, no), the clinical presentation if the case was confirmed (I have not passed the COVID-19 infection, asymptomatic, mild symptoms (sore throat, congestion, headache…), severe symptoms (significant dyspnea, confusion, asthenia, cyanosis…), living with the patient (severe dyspnea, confusion, asthenia, cyanosis…), living with the patient (yes, no), cohabitation with persons at risk for COVID (yes, no), COVID-19 infection of a family member/close relative (yes, no) and country of birth.

Data were collected during visits to classrooms. The purpose of the study was explained to the students as well as that their anonymity was assured. Those students who wished to take part did so over a period of 10–15 min of classroom time using the Google Forms^®^ online platform. Prior to completing the form, the students filled in the informed consent in the online form itself, after having received all the information and being asked all the appropriate questions. The data collected were then transferred to a Microsoft Excel 365^®^ database.

In relation to the statistical analysis, a general description of the main variables of the study was carried out. The normal distribution of the variables was tested using the Kolmorov–Smirnov test. Quantitative variables were described by the mean and standard deviation and the qualitative variables by their frequency distribution and percentages. For comparison between variables (*p* < 0.05), the chi-square test was used for qualitative variables. The corresponding CI (95%) was calculated. Multivariate analysis was performed, using forward stepwise binary logistic regression, with calculation of adjusted ORs and their corresponding 95% confidence intervals. Statistical analysis of the results was performed with the Statistical Package for the Social Sciences (SPSS) version 25 for Windows, EE.UU (IBM©).

The study was evaluated by the Research Committee of the Faculty of Nursing, Physiotherapy and Podiatry of the Universidad Complutense de Madrid and by the Research Ethics Committee of the Hospital Universitario Clínico San Carlos.

The study was conducted in accordance with the basic principles of the Declaration of Helsinki (Fortaleza 2013). The data were incorporated into a database in which no reference to the identity of the subjects was recorded. The processing, communication and transfer of their data was done in accordance with the provisions of the RGPD (EU Regulation) 2016/679 of the European Parliament and of the Council of 27 April 2016 and the LOPDGDD 3/2018, of 5 December.

## 3. Results

The starting point was a total population of 3222 students. The final sample was 1894 participants (58.8%). Of the 1582 students who were sent the Q1 questionnaire, 934 answered (49.3%) and, of the 1640 students who received the Q2, 960 (50.7%) answered. Of the sample, 80.9% were female, with a mean age of 21.44 years (SD 5.76 years), slightly lower in Q1, 21.30 years (SD 5.34 years) and slightly higher in Q2, 21.58 years (SD 6.13 years); 62.1% were younger than 21 years and 70.5% studied nursing (Table 1).

### 3.1. Influenza Vaccination Coverage

In the Q1 questionnaire, the coverage reported by students in the 2019/2020 campaign was 26.7%, compared to the 35.0% reported by students in the Q2 questionnaire (2021/2022 season), the difference being statistically significant (*p* < 0.001). 

Table 2 shows the change in influenza vaccination coverage between Q1 and Q2 in relation to sex, age, degree studied and year of the course. Coverage increased significantly (*p* < 0.05) in students of both sexes, in those aged 21 years or older, in those studying nursing and podiatry degrees and in the first and fourth year courses.

In the students who participated in Q2, when analyzing the influence of the variables related to COVID-19 on the flu vaccination coverage of the students, it was observed that the coverage was higher in students vaccinated against COVID (35.5% vs. 14.3%) and in those who lived with persons at risk (42.1% vs. 30.5%) (Table 3).

Table 4 shows the results of the multivariate analysis, identifying the independent variables that influenced influenza vaccination coverage. Being a fourth year student was the variable that had the greatest positive influence on influenza coverage, as well as being a nursing student. Likewise, being in the second year and living with a person at risk also increased coverage.

### 3.2. Vaccination Coverage against SARS-CoV-2

Vaccination coverage against SARS-CoV-2 of the students participating in Q2 was 97.8%. Of the students, 25.6% were COVID cases, with 97.5% of them having asymptomatic or mild disease. Some 45.4% had a close acquaintance with COVID infection and 54.6% lived with a person at risk (Table 5). 

Of the students who were positive for COVID-19, 62.7% of the asymptomatic students were under 21 years of age, 62.7% were in nursing school and 94% were vaccinated against SARS-CoV-2. Of those with mild disease, 64.5% were under 21 years of age, 79.1% were nurses and 94.2% were vaccinated against SARS-CoV-2. Of those with severe disease, 83.3% were over 21 years of age, 83.3% were nurses and 100% were vaccinated against SARS-CoV-2, with a statistically significant association in all cases (Table 6). 

Table 7 shows that the only two variables that influenced COVID vaccination coverage in the multivariate analysis were “having passed COVID infection” and “having been vaccinated against influenza”. 

## 4. Discussion

### 4.1. Influenza Vaccination Coverage

Adherence to influenza vaccination has increased among health sciences students from 26.7% in the 2019–2020 season to 35% in the 2021–2022 season, coinciding with the COVID-19 pandemic. These data, despite being far from the target of 75–80%, which both national agencies such as the Ministry of Health [14] and international agencies such as the World Health Organization [15] and the European Commission [16] recommend, represent an increase in coverage that we can consider of some relevance, taking into account the difficulty in changing student behavior [17]. This increase in participation could be due to the pandemic, which made the students aware of the situations experienced and the fact that the prognosis of COVID worsens if it is associated with influenza (increasing the risk of death considerably [18]), and this has generated positive attitudes of the students towards the influenza vaccine as a system of self-protection and to avoid becoming possible vectors of the disease in their practices in health centers. The fact that the symptoms of influenza and those of COVID-19 are similar and protection against influenza through vaccination can reduce the number of misdiagnoses may also have had an influence. 

If we compare the coverage achieved with that obtained in another study conducted in Spain in 2017 with nursing students, we see that it is considerably higher, since the coverage achieved in that study was 15.2% [19]. Internationally, there are studies that have identified higher coverages, such as the one conducted in Poland among medical students where influenza coverages reached up to 44.4% [20]; there was another one conducted in the United Kingdom among nursing students where coverage in 2019 reached up to 47.8% [21], one in California among public health students where coverage was 43.0% (2018) [22] and 52.4% in another study conducted among nursing students in Canada in 2019 [23]. However some studies present similar coverages of 38.4% [24], 36.7% [21] and 27.3% [25], or lower ones of 27.3% [26] and 16.5% [27]. 

When analyzing the change in coverage in a disaggregated manner, an increase in coverage was observed in students of both sexes and in older students, perhaps associated with a greater perception of the seriousness associated with suffering influenza and SARS-CoV-2 infection. Similar results were found in a study where coverage was higher among older master’s students compared to undergraduates [26] and a study conducted in the United States [22]. Increased coverage was also observed in students with nursing and podiatry degrees. Different results were found in a study in Canada, where nurses had lower coverage than other health professionals [23]. Perhaps in our study, nursing students, as they do their internships in health centers, are those who are at greater risk of exposure to immunopreventable diseases such as influenza and also receive more hours of training related to vaccines, as they are the protagonists of the vaccination act. Coverage was also higher in the first and fourth grades. Similar results were found in a study conducted in the United States [22], with the first year students having just joined the university community and showing greater adherence to public health guidelines, and the fourth year students showing greater awareness due to the greater number of hours spent in the hospital. Those students who lived with people at risk were more likely to be vaccinated because they were more concerned about the safety of such contacts, such as family members or friends [28]. Several studies have investigated how an experience with COVID-19 may affect vaccination intention [29,30,31].

### 4.2. Vaccination Coverage against SARS-CoV-2

Vaccination coverage against SARS-CoV-2 among health sciences students was very high (97.8%). Similar results were found in a study done in Texas among medical students where coverage was 91.8% [32]. However, lower coverages were also found in other studies conducted in Slovakia and Japan among medical students where the coverages were 71.7% [33] and 89.1% [34], respectively. Perhaps the differences are due to greater reliance on the vaccine over time, the educational environment or differences in the type of vaccine available in each country [35,36,37]. 

Being vaccinated against influenza was a predictor that favored vaccination against SARS-CoV-2, results similar to those of other studies such as the one carried out in Slovakia [33], perhaps, as previously noted, due to the protective effect of influenza vaccination against COVID-19. Despite different studies [38,39], there is no conclusive evidence that influenza vaccination has a protective effect on COVID-19, risk of infection or severity of illness. The greater propensity to be vaccinated against COVID-19 in those who were vaccinated against influenza may be related to both greater attention to recommendations and greater concern for one’s own health, resulting in greater compliance. Whatever the reason, this fact has been observed by different studies [40].

On the contrary, having been a positive case of COVID-19 was a negative predictor, perhaps influenced by the belief that natural immunity protects against COVID-19 infection [34] or by the time that must elapse between infection and the administration of the corresponding dose of vaccine. 

Regarding the severity of COVID-19, a very small percentage presented a severe picture in both men and women. This may have been due to the very high vaccination rate. Students in the age group above 21 years presented more severe pictures, as well as those in the Nursing degree. This could be due to the fact that age is a risk factor for the prognosis and evolution of the disease and also that nursing students were in more direct contact with cases of COVID-19 due to their internships in hospitals and health centers. Those vaccinated against SARS-CoV-2 presented mostly mild symptoms while those not vaccinated generally reported not having passed the COVID-19 infection, and only 4 students reported being asymptomatic during the course of the infection, while 10 students reported mild clinical symptoms. This may have been due to the fact that the questionnaire did not ask whether they had had the disease before or after vaccination against SARS-CoV-2. Similar results were found in a study done in Saudi Arabia [41], and another in Helsinki [42]. 

As a practical implication of this study, activities should be implemented in some subjects to encourage increased influenza vaccine coverage and reinforce vaccination against SARS-CoV-2.

This study has some limitations, it is a cross-sectional study using a self-administered questionnaire and the accuracy of the results is based on the responses of the participants which could have been inaccurate. A possible non-response bias, typical in this kind of research, could also be observed. The proportion of non-responders was around 50%, which could have influenced the vaccination coverage of the study. Perhaps they were not so high, because non-responders could be those not vaccinated against both influenza and SARS-CoV-2. Perhaps the lack of confidence in participating in online surveys (despite having been informed of the confidentiality and anonymity of the research), and the fact that the questionnaire did not include variables that would allow identification of the student, justifies this response rate. If the response rate had been very high, perhaps coverage would not have increased as much, which might have influenced the hypothesis. Another limitation was that the students were not the same, only about half of the Q2 participants coincided, which could influence the final coverage. It would have been necessary to collect this variable in order to analyze its influence on the final coverage. Another limitation is that this study was carried out in only one university. It is necessary to carry out similar studies in the rest of the universities to determine whether the data obtained are similar in the rest of Spain, which is foreseeable given the similarity of the training programs and the means available.

## 5. Conclusions

Coverage of the influenza vaccine in health sciences students increased from 2019–2020 to 2021–2022, being higher in the age group older than 21 years, who studied nursing, were in their first and fourth year and lived with people at risk.

Coverage of the SARS-CoV-2 vaccine in health sciences students was very high, especially in those vaccinated against influenza.

The importance of the high SARS-CoV-2 vaccination coverage rate lies in the fact that it reduces the risk of infection in students and decreases the probability of being a source of infection for patients, their classmates and family members.

The severity of the symptomatology presented by the health sciences students was generally mild.

## Figures and Tables

**Table 1 vaccines-10-00159-t001:** Characteristics of the sample for Q1 and Q2.

Variables		Total (Q1,Q2) n (%)	Q1 n (%)	Q2 n (%)
Sex	Female	1532 (80.9)	743 (48.5)	789 (51.5)
	Male	362 (19.1)	191 (52.8)	171 (47.2)
Aggregate age	<21 years old	1176 (62.1)	591 (50.3)	585 (49.7)
	≥21 years	718 (37.9)	343 (47.8)	375 (52.2)
Degree	Nursing	1336 (70.5)	624 (46.7)	712 (53.3)
	Physiotherapy	266 (14.0)	162(60.9)	104 (39.1)
	Podiatry	292 (15.4)	148(50.7)	144 (49.3)
Grade	1	547 (28.9)	292 (53.4)	255 (46.6)
	2	509 (26.9)	300 (58.9)	209 (41.1)
	3	431 (22.8)	178 (41.3)	253 (58.7)
	4	407 (21.5)	164 (40.3)	243 (59.7)

**Table 2 vaccines-10-00159-t002:** Flu vaccination coverage by sex, age, degree and course in Q1 and Q2.

Variables	Categories	Q1 n (%)	Q2 n (%)	*p*-Value
Sex	Female	202 (27.2)	277 (35.1)	*p* < 0.0001
	Male	48 (25.1)	59 (34.7)	0.031
Age	<21 years	155 (26.2)	175 (30.0)	0.087
	≥21 years	95 (27.7)	161 (42.9)	*p* < 0.0001
Degree	Nursing	203 (32.5)	278 (39.0)	0.008
	Physiotherapy	29 (17.9)	26 (25.0)	0.108
	Chiropody	18 (12.2)	32 (22.4)	0.015
Course year for all degrees taken together	1st	38 (13.0)	62 (24.3)	*p* < 0.0001
	2nd	121 (40.3)	84 (40.2)	0.524
	3rd	56 (31.5)	72 (28.6)	0.295
	4th	35 (21.3)	118(48.6)	*p* < 0.0001

**Table 3 vaccines-10-00159-t003:** Influenza vaccine coverage according to factors related to COVID infection (in students participating in Q2).

Variables		Q2	*p*-Value
		n (%)	
Country of birth	Spain	305 (35.2)	0.717
	Other	31 (33.3)	
Has been vaccinated against SARS-CoV-2	Yes	333 (35.5)	0.044
	No	3 (14.3)	
Has been diagnosed with COVID-19?	Yes	81 (35.5)	0.859
	No	255 (34.9)	
Type of symptomatology after being diagnosed with COVID	I have not passed COVID infection 19	248 (34.7)	0.813
	Asymptomatic	22 (32.8)	
	Mild case	63 (36.6)	
	Severe	3 (50.0)	
Has anyone you know close to you been COVID-19	Yes	152 (34.9)	0.956
	No	184 (35.1)	
Do you live with people at risk?	Yes	159 (42.1)	*p* < 0.001
	No	177 (30.5)	

**Table 4 vaccines-10-00159-t004:** Multivariate model using logistic regression: variables influencing influenza vaccination coverage.

Dependent Variable	Independent Variables	Coefficients B	OR (95%CI)	*p* Value
Vaccination to influenza	Course Year 4th (vs. 1°, 2° y 3°)	0.929	2.53 (1.83–3.50)	*p* < 0.001
	Nursing (vs. Physiotherapy and Podiatry)	0.682	1.98 (1.41–2.77)	*p* < 0.001
	Course Year 2rd (vs. 1°, 3° y 4°)	0.663	1.94 (1.37–2.74)	*p* < 0.001
	Cohabitation with at-risk patient (vs. no cohabitation)	0.514	1.67 (1.27–2.21)	*p* < 0.001
	Constant	−1756	0.063	0.007

**Table 5 vaccines-10-00159-t005:** Characteristics of the Q2 student population in relation to COVID infection 19.

Variables		Q2 n (%)
Has been vaccinated against SARS-CoV-2	Yes	939 (97.8)
	No	21 (2.2)
Has been diagnosed with COVID-19?	Yes	245 (25.6)
	No	715 (74.4)
Type of symptomatology after being diagnosed with COVID	Asymptomatic	67 (27.3)
	Mild case	172 (70.2)
	Severe	6 (2.5)
Has anyone close to you been a COVID-19 case?	Yes	436 (45.4)
	No	524 (54.6)
Do you live with people at risk?	Yes	379 (39.5)
	No	581 (60.5)

**Table 6 vaccines-10-00159-t006:** Severity of COVID-19 as a function of different factors.

Variables		Asymptomatic	Mild Clinical	Sever Clinical	*p*-Value
Sex	Female	46 (68.7)	148 (86.0)	5 (83.3)	0.144
	Male	21 (31.3)	24 (14.0)	1 (16.7)	
Age	<21 years old	42 (62.7)	111 (64.5)	1 (16.7)	0.017
	≥21 years old	25 (37.3)	61 (35.5)	5 (83.3)	
Country of birth	Spain	57 (85.1)	149 (86.6)	6 (100.0)	0.079
	Others	10 (14.9)	23 (13.4)	0 (0.0)	
Grade	Nursing	42 (62.7)	136 (79.1)	5 (83.3)	0.016
	Physiotherapy	16 (23.9)	16 (9.3)	0 (0.0)	
	Podiatry	9 (13.4)	20 (11.6)	1 (16.7)	
Course	1	17 (25.4)	57 (33.1)	3 (50.0)	0.069
	2	11 (16.4)	26 (15.1)	0 (0.0)	
	3	23 (34.3)	50 (29.1)	2 (33.3)	
	4	16 (23.9)	39 (22.7)	1 (16.7)	
Influenza vaccinated Q2	Yes	22 (32.8)	63 (36.6)	3 (50.0)	0.813
	No	45 (67.2)	109 (63.4)	3 (50.0)	
COVID-19 vaccinated	Yes	63 (94.0)	162 (94.2)	6 (100.0)	*p* < 0.001
	No	4 (6.0)	10 (5.8)	0 (0.0)	

**Table 7 vaccines-10-00159-t007:** Multivariate model using logistic regression: variables influencing vaccination coverage against SARS-CoV-2.

Dependent Variable	Independent Variables	Coefficients B	OR (95%CI)	*p* Value
Vaccination to SARS-CoV-2	Have passed the COVID infection	−1.329	0.27 (0.12–0.65)	0.003
	Have been vaccinated against influenza	1.21	3.36 (0.98–11.54)	0.054
	Constant	3.994	54.28	*p* < 0.001

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
