# Peer review of "Vaccine Coverage against Influenza and SARS-CoV-2 in Health Sciences Students during COVID-19 Pandemic in Spain"

_vaccines, 2022, doi:10.3390/vaccines10020159_

Round 1

Reviewer 1 Report

In this interesting manuscript, the authors aimed to to evaluate the change in influenza vaccination coverage during the 2019/2020 and 2021/2022 campaigns, to analyze the current vaccination coverage against SARS-CoV-2, and to identify factors associated with vaccination, in both cases, in health sciences students.

This is an important topic as there has been little influenza activity around the world since April 2020 and achieving high vaccination coverage rates will be important when influenza will circulate.

Some comments are made below:

  • If I understand it correctly, changes in influenza vaccination coverage among students are referred to the last two seasons (2019/2020 and 2020/2021). I suggest changing 2021/2022 (current influenza vaccination season) with 2020/21 – OR, if this is the case – better explaining that the authors are comparing coverage rates (2019/20) with intention to get vaccinated (21/22). It is quite important to resolve this misunderstanding. 
  • You correctly write that “as a consequence of this massive campaign launched worldwide, there has been a change in the general confidence in vaccines and the willingness to receive the next dose of influenza vaccine, increasing from 53% to 60%. […] However, it is unclear whether this increased willingness will translate into higher coverage”. As a matter of fact, the last winter (2020/2021) showed indeed a remarkable increase in influenza vaccination coverage rates in different population groups in many European Countries [1] including Spain. This is particularly important in a period in which there is little influenza activity. I suggest briefly mentioning this important fact in the introduction.

[1] Del Riccio M, Lina B, Caini S, Staadegaard L, Wiegersma S, Kynčl J, Combadière B, MacIntyre CR, Paget J. Letter to the editor: Increase of influenza vaccination coverage rates during the COVID-19 pandemic and implications for the upcoming influenza season in northern hemisphere countries and Australia. Euro Surveill. 2021 Dec;26(50). doi: 10.2807/1560-7917.ES.2021.26.50.2101143. PMID: 34915972.

  • Why were students enrolled from the Nursing, Physiotherapy and Podiatry faculties only?
  • In the Results section, you write “Of the 1,582 students who were sent the Q1 questionnaire, 934 128 answered (49.3%) and of the 1,640 students who answered the Q2 questionnaire, 960 129 answered (50.7%)”. If I understand correctly, you mean that 1640 were sent the Q2 questionnaire. I suggest rephrasing.
  • I suggest hypothesise and explaining how the non-response bias may have affected your results.

The discussion is interesting and nicely written. Here some points that can help further improve it:

  • You correctly mention that “being vaccinated against influenza was a predictor that favoured vaccination against SARS-CoV-2, results similar to those of other studies such as the one carried out in Slovakia, perhaps, as previously noted, due to the protective effect of influenza vaccination against COVID-19”. Despite being noted by different studies [2-3], there is no conclusive evidence that influenza vaccination has a protective effect on COVID-19 risk of infection or severity of disease. I suggest explaining this, to avoid the reader think that this is established knowledge. The higher propension to get vaccinated against covid-19 for those who vaccinated against influenza may be related to both a higher attention paid to the recommendations and a higher preoccupation for one’s own health, resulting in higher compliance. Whatever the reason, this fact has been observed by different studies I suggest reporting, if you think they fit your discussion [4].

[2] Jehi, L.; Ji, X.; Milinovich, A.; Erzurum, S.; Merlino, A.; Gordon, S.; Young, J.B.; Kattan, M.W. Development and validation of a model for individualized prediction of hospitalization risk in 4536 patients with COVID-19. PLoS ONE 2020, 15, e0237419

[3] Ragni, P.; Marino, M.; Formisano, D.; Bisaccia, E.; Scaltriti, S.; Bedeschi, E.; Grilli, R. Association between Exposure to Influenza Vaccination and COVID-19 Diagnosis and Outcomes. Vaccines 2020, 8, 675. https://doi.org/10.3390/vaccines8040675

[4] Sallam M, Dababseh D, Eid H, Al-Mahzoum K, Al-Haidar A, Taim D, Yaseen A, Ababneh NA, Bakri FG, Mahafzah A. High Rates of COVID-19 Vaccine Hesitancy and Its Association with Conspiracy Beliefs: A Study in Jordan and Kuwait among Other Arab Countries. Vaccines (Basel). 2021 Jan 12;9(1):42. doi: 10.3390/vaccines9010042.

Author Response

REVIEWER 1. Modifications have been underlined in yellow.
- Vaccine coverages are studied in the 2019/2020 season, before the pandemic, and the 2021/2022 season in the midst of the pandemic. All dates in the document have been reviewed for any errors in the document. In addition to the specific objectives that state: "The objectives of this study were to evaluate the change in influenza vaccination coverage during the 2019/2020 and 2021/2022 campaigns, to analyze current vaccination coverage against SARS-CoV-2, and to identify factors associated with vaccination, in both cases, in students of health sciences", where explicit reference is made to the two campaigns.
In addition, the wording of Materials and Methods was modified, clarifying the period of data collection, which could have given rise to some confusion. Data collection was carried out between January 15-30, 2020 (Q1), belonging to the 2019/2020 campaign and during October to November 2021 (Q2), where data were collected from the campaign that was being developed and therefore from the 2021/2022. Most of the students had already been vaccinated and those missing had an appointment for vaccination. The latter were counted as having been vaccinated. In no case was reference made to the intention to vaccinate.
- It has been included in the introduction as suggested by the reviewer "It is also important to highlight the increase in influenza vaccination coverage in the 2020/2021 season despite, on the one hand, low influenza activity or the existence of a problem of underdiagnosis, since it is thought that COVID-19 cases could not suffer simultaneously from both pathologies"
- Only the Faculty of Nursing, Physiotherapy and Podiatry were chosen, firstly because of the accessibility of the research group and secondly because, depending on the results, it could be extended to the rest of the students of the Complutense University and other Universities in Madrid. A sentence justifying this is included in Material and Methods. "This faculty was the only one chosen because of the accessibility of the researchers and the management of resources, pending the results obtained."
- The results sentence is rephrased as follows "and of the 1,640 students who received the Q2, 960 (50.7%) answered it".
- The initial hypothesis is formulated in the Material and Methods section: "As an initial hypothesis, it was assumed that vaccination coverage would increase considerably as a consequence of the pandemic".
- It is added in the last paragraph of the discussion, limitations of the study: "The nonresponse bias, which was around 50%, could have influenced the vaccination coverages of the study. Perhaps they were not as high, because the nonresponders could be those who were not vaccinated against both influenza and SARS-CoV-2." "Influencing the initial hypothesis, where if the response rate had been very high, perhaps the coverages would not have increased as much."
- The four references suggested by the reviewer have been introduced.
- It is added in the discussion that there is no conclusive evidence that influenza vaccination has a protective effect on the risk of COVID-19 infection or severity of disease "Despite
different studies [35,36], there is no conclusive evidence that influenza vaccination has a
protective effect on COVID-19, risk of infection, or severity of disease. The greater propensity
to be vaccinated against COVID-19 in those who were vaccinated against influenza may be
related to both greater attention to recommendations and greater concern for one's own
health, resulting in greater compliance. Whatever the reason, this fact has been observed by
different studies [37]."

Reviewer 2 Report

This is an interesting study on significant public health problems. Students of medical faculties are an important research group in the case of research on attitudes towards vaccination. 

Some changes should be applied:

  1. The title should include information on where this study was carried out (Spain)
  2. Lines 57-58 should be revised ("Currently there is still no clear treatment for the virus"; treatment of the disease (COVID-19) rather than virus)
  3. The Authors should consider adding 1 paragraph on vaccination policy in Spain (access to vaccination point; rules of referral to vaccination point; cost for students) - background information on vaccination policy will be helpful for the international readers 
  4. The authors should clearly define sampling methods and the population included in this study (e.g., whole student groups? or only selected ones?).
  5. The study design is a little bit confusing - please clearly define the logical structure of the text (e.g form of the questionnaire - paper-based or online, etc.)
  6. Please consider the division of the Material and Methods section into subsections (e.g., study design and population; study questionnaire; statistical analysis)
  7. Results
    1. please clearly define the response rate 
    2. please clearly define why the age was aggregated? 
    3. most of the respondents were females and this issue should be addressed in the discussion section
    4. English language used in Tables (e.g description of the variables in Table 3 should be revised) 
  8. Please provide practical implications of this study

Author Response

REVIEWER 2. Modifications have been underlined in this color.
- Add "in Spain" in the title.
- The text in lines 57-58 is modified as follows: "Currently there is still no clear treatment of the disease". Treatment of the virus is eliminated.
- The following paragraph related to the vaccination policy for university students in Spain is added in the introduction: "In Spain, students of Health Sciences are recommended at the University to be vaccinated against influenza and currently against SARS-CoV-2 before starting their hospital internships. They are vaccinated at their health centers, by appointment requested by them. The administration of these two vaccinations is free of charge for the students.
- In the section on material and methods, a sentence is included in which it is stated that all students were included, without any type of sampling. "All students of the Faculty were included, without any type of sampling".
- The data collection process has been better described in the material and methods section.
- The variables in Table 3 have been reworded, as they could lead to misunderstandings when reading them.
- A paragraph has been added in the discussion before the paragraph on the limitations of the study recommending the promotion of vaccination against influenza and reinforcing vaccination against SARS-CoV-2: "a practical implication of this study, activities should be implemented in some subjects to encourage increased influenza vaccine coverage and reinforce vaccination against SARS-CoV-2".

Reviewer 3 Report

The manuscript investigates vaccination coverage against influenza and SARS-CoV-2 in health science students in Madrid. The topic is timely and interesting. A few comments to improve the manuscript.

Abstract

Abstract should describe the main findings and conclude with a brief discussion. In this abstract the main focus are univariable analyses and there is no conclusion.

Methods

Were the students the same in the two surveys, meaning that the same person may have filled out both questionnaires? If yes, this should be taken into account in the analysis and also in the discussion. If no, this concept should be mentioned in the limitations.

Page 2, line 97. Living with the patient (no, no, no, no). What does it mean?

How was vaccination coverage defined? One or two doses?

Multivariate is not multivariable. Be careful how you define your models.

Why a cut-off of 21 years old was chosen to categorize age? This should be added to the methods section.

Table 1. The % of sex are not correct. 743 females cannot be 48.5% while 191 males represent 52.8%. Please check the entire table.

Table 4. Does the analysis consider Q1 students, Q2 students or both? If both, why the variable differentiating the two surveys is not included? Probably it would have been better to include course year as a categorical variable with the first year as reference category.

Table 6. This data is off topic since the focus was vaccination coverage and not the infection symptoms. Also, grade is repeated twice.

Discussion

Page 8, line 235-238. Several studies have investigated how an experience with COVID-19 may affect vaccination intention and it rarely played a role. Suggested references worth including are:

  • Saied SM, et al. Vaccine hesitancy: Beliefs and barriers associated with COVID-19 vaccination among Egyptian medical students. J Med Virol. 2021 Jul;93(7):4280-4291. doi: 10.1002/jmv.26910.

  • Baccolini V, et al. COVID-19 Vaccine Hesitancy among Italian University Students: A Cross-Sectional Survey during the First Months of the Vaccination Campaign. Vaccines (Basel). 2021 Nov 7;9(11):1292

  • Tavolacci MP, et al. COVID-19 Vaccine Acceptance, Hesitancy, and Resistancy among University Students in France. Vaccines (Basel). 2021 Jun 15;9(6):654. doi: 10.3390/vaccines9060654.

The fact that that students with a past-infection of COVID-19 were associated with lower likelihood of vaccination coverage could also depend on how vaccination coverage was defined (one dose or two doses?). In many countries, for people with a past infection, only one dose was initially recommended and considered as sufficient to ensure immunity.

Author Response

REVIEWER 3. Modifications have been underlined in this color.
- Added conclusions in the abstract.
- Included in the methods section the observation that the students were not the same as those who responded in Q1, at least half coincided in Q2. This could have influenced the final result of the coverage. It would have been interesting to collect this variable to study its influence on coverage against influenza and SARS-CoV-2. "The students who participated were not the same. Those from two nursing, physiotherapy and podiatry courses coincided, the rest were different". In addition, another paragraph was included in the limitations of the study. "Another limitation was that the students were not the same, only about half of the Q2 participants coincided, which could influence the final coverage. It would have been necessary to collect this variable in order to analyze this influence on the final coverage."
- On page 2, line 97, the error in the variable "living with the patient (yes, no)" is corrected.
- Vaccination coverage against SARS-CoV-2 was measured when at the time of the questionnaire they had received both doses of vaccine or one dose and had passed the disease.
- The nomenclature of the multivariate study was revised, modifying any errors that might exist.
- It was added in the material and methods section that the cut-off age in the categorization of the variable age was 21 years. This value was chosen as the limit in the categorization of the variable age, as it was the value at which the number of students of this age and above decreased" (<21 years old, >=21 years; 21 years of age was chosen as the limit in the categorization of the variable age, as it was the value at which the number of students of this age and above decreased)".
- The data refer to the percentage that participated in both Q1 and Q2 within the same category, i.e. 743 females were 48.5% of the total number of females participating in Q1 and Q2. 191 men were 52.8 % of the total men who participated in Q1.
- In relation to the reviewer's suggestion to include the variable that differentiates the two surveys, it was decided to include the total without differentiating the year of the course, because we were looking for a general predictor for lower power and not by year of the course.
- To describe the severity of the signs and symptoms presented by the students when they were diagnosed with a COVID-19 case" is added to the last paragraph of the introduction where the objectives of the study are described. In table 6 correct "course by grade".
- A sentence as suggested by the reviewer and the 3 bibliographic citations are included. In addition, the references are included at the end of the article. "Several studies have investigated how an experience with COVID-19 may affect vaccination intention [29-31]."

Round 2

Reviewer 1 Report

In the revised version, the manuscript has improved in readability and overall quality. The authors made significant efforts to address all the comments.

Particularly, the purpose has been made more clear, the literature review updated to be current, and the way to pursue the research aim has been clearly explained.

The data analysis strategy adopted appears apt to answer the research question and the results are described correctly and discussed properly.

Reviewer 2 Report

The Authors addressed all the comments.

Reviewer 3 Report

None